# Roles of Early Mobilization Program in Preventing Muscle Weakness and Decreasing Psychiatric Disorders in Patients with Coronavirus Disease 2019 Pneumonia: A Retrospective Observational Cohort Study

**DOI:** 10.3390/jcm10132941

**Published:** 2021-06-30

**Authors:** Toru Kotani, Mizuki Sugiyama, Fumika Matsuzaki, Kota Kubodera, Jin Saito, Mika Kaneki, Atsuko Shono, Hiroko Maruo, Maiko Mori, Shin Ohta, Fumihito Kasai

**Affiliations:** 1Department of Intensive Care Medicine, Showa University School of Medicine, Tokyo 142-8666, Japan; atsukos@med.showa-u.ac.jp (A.S.); katohiroko@med.showa-u.ac.jp (H.M.); morimai@med.showa-u.ac.jp (M.M.); 2Department of Rehabilitation Medicine, Showa University School of Medicine, Tokyo 142-8666, Japan; sugiyama-m@med.showa-u.ac.jp (M.S.); fumihito@med.showa-u.ac.jp (F.K.); 3Rehabilitation Center, Showa University Hospital, Tokyo 142-8666, Japan; matsu.fumi@cmed.showa-u.ac.jp (F.M.); k.kubodera@cmed.showa-u.ac.jp (K.K.); 4Graduate School of Nursing and Rehabilitation Sciences, Showa University, Yokohama 226-8555, Japan; saito-jin@cmed.showa-u.ac.jp; 5Department of Nutrition, Showa University Hospital, Tokyo 142-8666, Japan; mika-kane@cmed.showa-u.ac.jp; 6Department of Internal Medicine, Showa University School of Medicine, Tokyo 142-8666, Japan; shinohta@med.showa-u.ac.jp

**Keywords:** postintensive care syndrome, coronavirus disease 2019, acute respiratory failure

## Abstract

Although many coronavirus 2019 patients have experienced persistent symptoms and a long-term decline in quality of life after discharge, the details of these persistent symptoms and the effect of early rehabilitation are still unclear. We conducted a single-center, retrospective observational study to investigate the prevalence of persistent symptoms three months after discharge from the intensive care unit by checking the medical records. All patients received an early mobilization program. Four out of 13 patients (31%) had postintensive care syndrome. No patients had muscle weakness, and 11 patients (85%) returned to their previous work. However, psychiatric disorder, such as anxiety (23%) and posttraumatic stress disorder (15%), were observed. Eleven patients claimed persistent symptoms, including fatigue and numbness in the extremities. Our results suggest that the implementation of an early rehabilitation program plays some role in preventing muscle weakness and that decreasing psychiatric disorders should be a next target of patient care in the intensive care unit.

## 1. Introduction

The worldwide spread of SARS-CoV-2 has caused an unprecedented pandemic of coronavirus disease 2019 (COVID-19) and has had a great impact on intensive care medicine. Numerous studies have gradually established treatments for this disease. However, it has also been reported that many patients experience persistent symptoms and a long-term decline in quality of life after discharge [1,2,3]. Although various sequelae have been reported, it has not been fully clarified whether these sequelae are caused by COVID-19 or are related to intensive care. The effect of early rehabilitation on COVID-19 patients is still under investigation.

The concept of postintensive care syndrome (PICS) was proposed in 2010. PICS includes motor dysfunction and mental and cognitive dysfunction that occurs and worsens during or after the intensive care. PICS affects not only the patient’s long-term prognosis and quality of life (QOL) but also the psychological conditions of the patient’s family members [4]. PICS is expected to be a major issue in the era of COVID-19, and the establishment of preventive measures through the surveillance of current situations is warranted [5,6]. Although evidence-based and supportive care in the ICU, such as the ABCDEF bundle, is expected to decrease PICS, a point prevalence study showed low implementation of the bundle [7]. The details of the long-term prognosis and PICS after COVID-19 infection have not been investigated in Japan. 

The purpose of this study was to investigate the incidence and clinical features of PICS and the impact on the quality of daily living after discharge in COVID-19 patients in our cohort. We also observed an association between early rehabilitation and PICS in the intensive care unit (ICU). 

## 2. Subjects and Methods

A single-center, retrospective observational study was performed, and data were collected by searching medical records. Of the patients who were admitted to the ICU of Showa University Hospital from 1 February 2020 to 31 January 2021, those that met the following three inclusion criteria were targeted for an investigation of PICS: (1) patients who were diagnosed with COVID-19 pneumonia; (2) patients who were admitted to the ICU and stayed more than 48 h for the treatment of COVID-19 pneumonia; and (3) patients who survived and were discharged from the ICU more than three months before the survey. Patients who met any of the following criteria were excluded: (1) patients who did not provide informed consent in a face-to-face explanation of the study and (2) patients who were not followed up after discharge because they did not attend outpatient visits. This study was approved by the local medical ethics committee of Showa University Hospital (No. 3350), and the requirement for written informed consent was waived due to the retrospective study design. 

The diagnosis of COVID-19 was made on the basis of a positive result in polymerase chain reaction testing, the presence of clinical symptoms of pneumonia, and positive findings on chest radiography. When the patient was placed on mechanical ventilation, dexmedetomidine was used as a first choice with fentanyl, followed by propofol administration. Sedation was assessed using the Richmond Agitation-Sedation Scale. Midazolam was the last choice because of its higher association with delirium. Neuromuscular blockade was used when prone positioning was required or strong inspiratory effort could not be controlled using maximal doses of sedatives. Delirium was monitored, diagnosed, and recorded by the trained ICU nurses and/or intensivists every 6 h or when needed using the Confusion Assessment Method for the ICU or Intensive Care Delirium Screening Checklist, as appropriate. An early mobilization (EM) program was ordered by the intensivist in charge and started by the nurses within 48 h of ICU admission. The EM program was implemented according to the protocol that was published in a previous study [8], with modification. Briefly, the EM program consists of 5 stepwise levels: passive range of motion, head up 30 degrees, sitting on the edge of bed, active transfer to wheelchair, and standing with assistance. If achieving the level without any unfavorable event, the patient proceeds to the next higher level. EM is carried out by either a physician, nurse, physical therapist, or occupational therapist. Each session is 20 min or more. 

A retrospective chart review was performed, and the following data were collected: baseline characteristics, details of treatments, the length of ICU and hospital stay, the incidence of delirium during the ICU stay, and sustained treatments after hospital discharge. Additionally, we investigated the daily living conditions of each patient in whom follow-up was possible up to 3 months after ICU discharge. The follow-up survey was carried out using both electronic medical records and face-to-face or telephone interviews with the patient using the same questionnaire of each assessment. Two doctors responsible for the interview had trained the use of the questionnaire to ensure the reliability of the question. The primary endpoint was the incidence of PICS 3 months after ICU discharge. PICS was defined using the following scores/scales: Medical Research Council (MRC)-sum score for motor dysfunction, Hospital Anxiety and Depression Scale (HADs) for both depression and anxiety, and the Impact of Event Scale-Revised (IES-R) as posttraumatic stress disorder (PTSD). PICS was diagnosed when the patient met any one of the following criteria: an MRC-sum score less than 48, an anxiety or depression score on the HADS of 8 or more, and an IES-R score of 25 or more. Mortality; physical symptoms; activities of daily living (ADLs) assessed by the Barthel Index (BI); health-related QOL, assessed by the EuroQol 5 dimensions 5-level (EQ-5D-5L, Japanese version); cognitive function, assessed by the Short-Memory Questionnaire (SMQ); and return-to-work status at 3 months after ICU discharge were investigated as secondary endpoints.

Statistical analyses were performed using JMP Pro 15 (SAS Institute Inc., Cary, NC, USA). The significance of the categorical variables was calculated using Fisher’s exact test. In the case of continuous variables, Mann–Whitney U tests were used for comparisons between patients with and without PICS. A *p* value less than 0.05 was considered statistically significant. Data are presented as the median (interquartile range, IQR) unless otherwise specified.

## 3. Results

In total, 585 patients were admitted due to suspected COVID-19 during the study period. Two hundred and seventy-one patients were diagnosed with COVID-19 pneumonia, and 35 patients were treated in the ICU to receive mechanical ventilation. Of those, 31 patients were intubated and mechanically ventilated. Seven patients were still being treated in the ICU and were excluded. Twenty-eight patients were discharged from the hospital, and 12 were excluded (eight died during the study and four died within 3 months of discharge). Three patients could not be followed up. Finally, 13 patients met the inclusion criteria (Figure 1). The average mortality rate in overall COVID-19 patients and severe COVID-19 patients admitted to the ICU were 4.0% and 22.9%, respectively.

All patients were males, aged 59 (48–74). At admission, the body mass index was 27.6 (22.9–30.3), and the arterial partial oxygen tension to inspired oxygen fraction (P/F) was 185 (122, 214) mmHg. The Charlson index was scored as 0 (6 patients), 1 (5 patients) or 2 (2 patients). The APACHE II score was 12.0 (8.0, 13.0). The treatments and outcomes for each patient are presented in Table 1 and summarized in Table 2. The duration of mechanical ventilation was 15 (7–57) days. The length of ICU and hospital stay were 12 (4–117) days and 21 (7–183) days, respectively. Delirium was observed in 6 patients. When starting EM session, the RASS was managed at -1 or higher. In the first four patients, the implementation of EM was delayed because information on the workflow dedicated to infection control and safety for COVID-19 was not provided. There was one patient (Patient 1) in whom the implementation of an early mobilization program was not described in the first 20 days in the medical record. In the remaining 12 patients, an EM program was provided for 28.0 (13.8, 46.0) hours, and 9 patients were able to perform mobilization activities within 48 h of ICU admission. No association was found between the implementation of an EM program and the occurrence of PICS. The mean body weight was 77.2 kg at ICU admission and decreased to 73.5 kg at ICU discharge. The maximal decrease in body weight after admission was 15%.

Table 3 shows the details of the physical and psychological follow-up at 3 months after ICU discharge. Psychological function of 13 patients was assessed by two pre-trained doctors (11 and 2, respectively). PICS was diagnosed in 4 patients (31%). Motor function measured by the MRC-sum score was well preserved or restored, which was consistent with full ADLs evaluated by the BI, leading to a high direct-home discharge rate (12 patients, 92%). Patient 9 was transferred to a rehabilitation hospital when investigated but was finally discharged home. Eleven patients (85%) could return to their previous work. Psychological function, however, was impaired in 4 patients: one was treated with oxygen therapy, and three were intubated (two of whom received ECMO). One patient showed cognitive impairment and sustained decreases in health-related QOL. The APACHE II score and Charlson index were not different between patients with PICS and those without PICS.

At the interview, 11 patients claimed they had persistent symptoms, including fatigue (6), numbness in the extremities (5), muscle weakness (5), hair loss (4), shortness of breath (3), and dysgeusia (3) (Figure 2), although muscle weakness was not determined by the testing.

## 4. Discussion

In our study four patients (31%) had PICS without muscle weakness after acute respiratory failure caused by COVID-19. Forty-six percent of patients had delirium, but only one patient had sustained cognitive dysfunction. A case series analysis in COVID-19 patients [9] reported a higher rate (75%) of delirium but a lower incidence of cognitive impairment and a higher 6 min walk distance at the 6 week follow-up than in non-COVID ARDS patients, quite similar to our findings. Previously, it was reported that the prevalence of PICS among ICU patients in Japan was 63.5%, and 32%, 15%, and 38% had physical, mental, and cognitive impairment, respectively, [10] indicating that the prevalence of psychological disorders was higher in that study. Dinglas and colleagues reported that 38% of patients with non-COVID-19 ARDS had significant muscle weakness at hospital discharge, sustained muscle weakness was found in 50%, and muscle weakness had an impact on the 5-year survival rate. A previous study conducted in the ARDS patients reported that EQ-5D after 6 months and 12 months from ICU discharge were 0.55 ± 0.37 and 0.58 ± 0.35, respectively [11]. Our study showed 0.80 ± 0.20, a higher recovery in a similar age cohort. These characteristics of sustained psychological symptoms and discrepancy with physical and cognitive symptoms might be associated with the pathophysiological characteristics of COVID-19.

One study reported that acute brain dysfunction was highly prevalent in critically ill patients with COVID-19, and the use of benzodiazepine and a lack of family visitation were identified as modifiable risk factors for delirium [12]. Our study showed a similar prevalence of delirium, although benzodiazepine infusion or bolus administration was restricted by the protocol in our ICU. In our study 62% of the patients received steroids, and 70% of mechanically ventilated patients were continuously paralyzed. These observations could be related to prolonged mechanical ventilation and the 4 kg body weight loss. However, muscle weakness was not found at 3 months after ICU discharge. These results could be explained by the fact that our early mobilization approach, which was standard and widely accepted, worked as expected and reduced or alleviated PICS, although the implementation of the program was delayed in four patients. A previous study showed that EM started soon after ICU admission reduced ICU-acquired weakness and delirium [13], but there was no clear evidence regarding the optimal timing for EM program implementation. Our results suggest that mobilization played some role in preventing or improving muscle weakness in critically ill COVID-19 patients, even if it was initiated lately. 

We found that 23% of the patients had anxiety and 15% had PTSD; these values were lower than those reported in a previous study in non-COVID acute lung injury patients [14], 22–24% and 38–44%, respectively, during a 2-year follow-up. All the questionnaires used in the study, the Japanese version of IES-R, HADS, and SMQ, have been validated. A single-center pilot study showed the efficacy of ICU diaries in reducing psychological morbidity following discharge [15]. In our clinical practice, an early mobilization program to restore muscle strength has been actively adopted, but mental and cognitive component prevention programs have been delayed. The result of the current study may reflect the lack of countermeasures for mental and cognitive decline.

Persistent psychiatric disorders, diagnosed when at least one of the three domains (anxiety, depression and PTSD) was observed, were reported in 66% of non-COVID ARDS survivors [16]. Carfi and colleagues [2] reported that 125 out of 143 patients (87.4%) who recovered from COVID-19 showed any of the symptoms of persistent psychiatric disorders 60 days after the onset of symptoms. Fifty-five percent of the patients had 3 or more symptoms. Our study showed a lower but similar incidence 3 months after ICU discharge. In the treatment of COVID-19, the patient was isolated in a negatively pressurized room for more than 2 weeks. These patients had no family visits until the ICU discharge. However, patients with persistent respiratory impairment do not always have psychiatric disorders. It was suggested that isolation from family members and society was related to the prevalence of sustained anxiety or PTSD.

Other persistent symptoms were similar to those in a previous report [17]. COVID-19 survivors mainly suffer from fatigue or muscle weakness, sleep difficulties, and anxiety or depression [3]. Among the sequelae, numbness remained in the upper or lower limbs in 5 patients (38.5%); prone positioning was applied in four of these five patients. Prone positioning was applied in 6 patients, and 4 of these patients had numbness as a sequela. In other patients, the dermatome of numbness was consistent with mononeuropathy or multiple mononeuropathy rather than polyneuropathy, meaning that it would be associated with root neuropathy. In the COVID era, prone positioning is often applied to maintain oxygenation. We also had to apply prone positioning for a maximum of 16 h in 5 patients on mechanical ventilation or ECMO, which led to excessive workload and inadequate care. Further studies are needed to investigate the relationship between numbness and prone positioning.

This study has several limitations. This was a single-center, observational study with a limited number of patients. Although we have prepared enough to prevent inter- and intra-investigator difference before using the questionnaire, the inter-investigator reliability was not evaluated in the current study. The study was not designed to compare the outcomes between groups with and without COVID-19 pneumonia. However, our study showed some role of the EM program in improving ICU-acquired impairment of QOL/ADLs.

## 5. Conclusions

PICS was observed in 31% of the COVID-19 patients and persisted in 23% of patients 3 months after ICU discharge. Motor function was relatively maintained by the implementation of an EM program, and the fact that many psychiatric disorders were sustained should be a next target of the care for the patients with COVID-19 in the ICU.

## Figures and Tables

**Figure 1 jcm-10-02941-f001:**
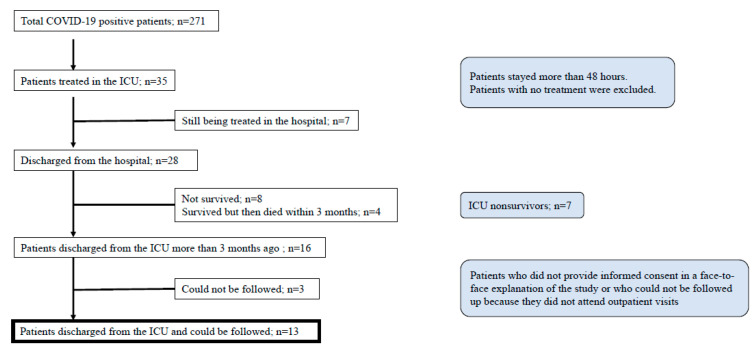
Patient flow chart.

**Figure 2 jcm-10-02941-f002:**
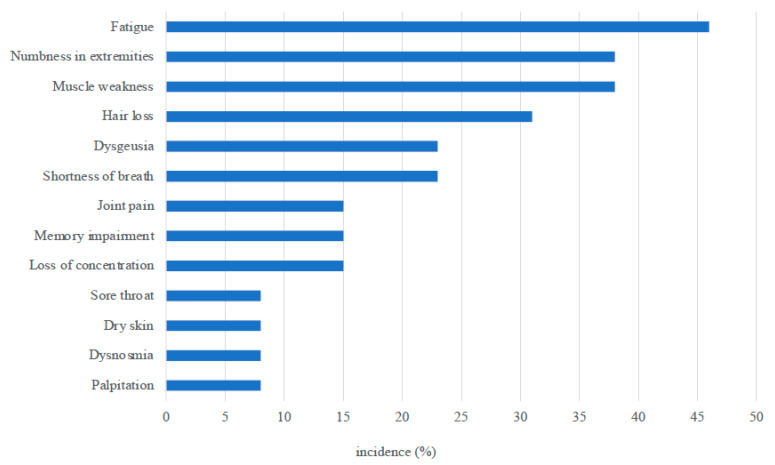
Persistent symptoms observed at 3 months after ICU discharge.

**Table 1 jcm-10-02941-t001:** Severity, comorbidity, treatments and outcomes of follow-up patients.

Patient	APACHE II Score at Admission	Charlson Index	Mechanical Ventilation	Duration of Ventilation (Days)	ECMO (Days)	Physical Restraint	Prone Positioning	In-Bed Cycle Ergometry (Days)	EMS (Days)	Admission to EM Implementation (Hours)	ICU LOS	Hospital LOS	Outcome
1	7	0	yes	21	10	yes	yes	NA	NA	NA	29	34	alive
2	12	0	yes	7	NA	yes	no	NA	NA	45	6	7	alive
3	13	1	yes	8	NA	yes	no	NA	NA	55	7	10	alive
4	7	1	yes	98	46	yes	yes	2	NA	98	117	183	alive
5	8	0	yes	8	NA	yes	no	NA	NA	49	11	21	alive
6	12	0	yes	7	NA	yes	no	NA	NA	31	12	21	alive
7	23	2	yes	23	NA	yes	yes	NA	NA	25	40	57	alive
8	8	0	no	NA	NA	no	no	NA	NA	2	8	14	alive
9	13	0	yes	57	19	yes	yes	12	2	7	65	118	alive
10	8	1	yes	35	NA	yes	yes	9	NA	5	44	64	alive
11	12	1	yes	8	NA	yes	yes	5	NA	16	11	15	alive
12	14	1	no	NA	NA	no	no	NA	NA	25	7	20	alive
13	12	2	no	NA	NA	no	no	NA	NA	45	4	10	alive

APACHE: acute physiological and chronic health evaluation, ECMO: extracorporeal membrane oxygenation, EMS: electrical muscle stimulation, EM: early mobilization, LOS: length of stay, NA: not applicable.

**Table 2 jcm-10-02941-t002:** Summary of treatments and outcomes.

Treatments	
Mechanical ventilation, *n* (%)	10 (77)
Prone position, *n* (%)	6 (46)
Veno-venous ECMO, *n* (%)	3 (23)
Use of continuous muscle paralysis, *n* (%)	7 (54)
Use of glucocorticoid, *n* (%)	8 (62)
Length of mechanical ventilation, median [IQR] (days)	15 (7–57)
Length of ECMO, median [IQR] (days)	19 (10–46)
Length of ICU stay, median [IQR] (days)	12 (4–117)
Length of hospital stay, median [IQR] (days)	21 (7–183)
Delirium during ICU stay, *n* (%)	6 (46)
Oxygen therapy at hospital discharge, *n* (%)	5 (39)
Direct-home discharge, *n* (%)	12 (92)

**Table 3 jcm-10-02941-t003:** Psychological assessment at 3 months after ICU discharge and long-term outcomes.

Patient	MRC-Sum Score	HADs Anxiety	HADs Depression	IES-R	PICS	BI	EQ-5D Mobility	EQ-5D Self-Care	EQ-5D Usual Activities	EQ-5D Pain/Discomfort	EQ-5D Anxiety/Depression	EQ-5D Calculation	SMQ (/46)	Return to Work	Home Discharge
1	60	0	0	1	no	100	1	1	1	1	1	0.9384	46	yes	yes
2	60	0	0	0	no	100	1	1	1	1	1	0.9384	46	yes	yes
3	60	0	0	0	no	100	1	1	1	2	1	0.8978	46	yes	yes
4	60	7	8	26	yes	90	4	4	4	4	4	0.1896	33	no	yes
5	60	0	1	6	no	100	1	1	1	2	1	0.8978	46	yes	yes
6	60	0	0	0	no	100	2	1	1	1	1	0.873	44	yes	yes
7	60	0	1	0	no	100	3	1	4	1	1	0.6708	46	yes	yes
8	60	1	8	6	yes	100	2	2	1	1	1	0.835	46	yes	yes
9	54	4	6	25	yes	95	3	2	3	2	1	0.6555	41	no	no
10	60	0	0	1	no	100	2	1	1	3	1	0.805	46	yes	yes
11	60	3	8	16	yes	100	1	1	1	3	1	0.8704	41	yes	yes
12	60	2	4	6	no	100	1	1	1	1	1	0.9384	42	yes	yes
13	60	0	1	2	no	100	1	1	1	1	1	0.9384	46	yes	yes

MRC: Medical Research Council, HADS: Hospital Anxiety and Depression Scale, IES-R: Impact of Event Scale-Revised, BI: Barthel Index, EQ-5D-5L: EuroQOL 5 dimensions 5-level, SMQ: Short-Memory Questionnaire.

## Data Availability

The data presented in this study are available on request from the corresponding author. The data are not publicly available due to privacy reason.

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
