# Peer review of "Roles of Early Mobilization Program in Preventing Muscle Weakness and Decreasing Psychiatric Disorders in Patients with Coronavirus Disease 2019 Pneumonia: A Retrospective Observational Cohort Study"

_jcm, 2021, doi:10.3390/jcm10132941_

Round 1

Reviewer 1 Report

Review Letters

Title

Current status of postintensive care syndrome in patients with severe acute respiratory failure caused by coronavirus disease 2019: a retrospective observational cohort study.

Dear Authors,

Please revise the following points in your manuscript.

Major review:

Overall, I think it must have been hard to experiment and write a thesis in a difficult environment. After reading the paper, three important issues to consider are presented below.

First, In the author's paper, EM is considered to be an important factor. However, without explaining this important factor in detail, the authors express only as "The EM program was implemented according to the protocol that was published in a previous study (8), with modification." This is thought to be difficult for readers to understand, so please explain in detail the EM program you performed in this study.

Second, Has the reliability of the questionnaires used by the authors been investigated?

Third, Data assessed by questionnaires that have not been tested for reliability are discrediting. Because of this, I wonder if the results as well as the discussion were expressed correctly.

Minor review:

Please, change the statistic symbol ‘P’ to italics (P).

Please, change the ‘n’ to italics (n).

All references should be formatted to fit the J. Clin. Med.

I hope my review helped you improve your manuscript.

Author Response

Major review:

Overall, I think it must have been hard to experiment and write a thesis in a difficult environment. After reading the paper, three important issues to consider are presented below.

First, In the author's paper, EM is considered to be an important factor. However, without explaining this important factor in detail, the authors express only as "The EM program was implemented according to the protocol that was published in a previous study (8), with modification." This is thought to be difficult for readers to understand, so please explain in detail the EM program you performed in this study.

Response: The authors appreciate your advice. We have added the details of the 5-stepwise program protocol and the procedure to carry out the program those can explain the details of our EM program. 

Second, Has the reliability of the questionnaires used by the authors been investigated?

Response: Thank you for the comments. The questionnaires used in the study, the Japanese version of IES-R, HADS, and SMQ, have been validated.

IES-R     DOI: 10.1097 / 00005053-200203000-00006 

HADS     DOI: 10.1111 / j.1600-0447.1983.tb09716.x

SMQ      DOI: https://doi.org/10.11477/mf.1406901279

Third, Data assessed by questionnaires that have not been tested for reliability are discrediting. Because of this, I wonder if the results as well as the discussion were expressed correctly.

Response: As answered above, the validation studies of all the questionnaires used in the study (the Japanese version of IES-R, HADS, and SMQ) have already been published. We have added the sentence in the Discussion section.  

Minor review:

Please, change the statistic symbol ‘P’ to italics (P).

Please, change the ‘n’ to italics (n).

All references should be formatted to fit the J. Clin. Med.

Response: We have corrected according to all these advises.

I hope my review helped you improve your manuscript.

Response: Thank you for the valuable comments to improve our manuscript.

Reviewer 2 Report

Dear Authors,

this is a well written manuscript reporting data of  a well designed, structured retrospective study.

Minor issues:

1. Please improve the quality of tables and figures.

2. The authors state that sedation depth was monitored with RASS - please provide data.

3. CAM-iCU and ICDSC was used to identify delirium - please provide details.

I have no further comments and wish the authors all the best for their future endeavors.

With best regards

Author Response

Minor issues:

  1. Please improve the quality of tables and figures.

Response: Thank you for the comment. We have improved the quality by using larger fonts.

  1. The authors state that sedation depth was monitored with RASS - please provide data.

Response: We have added the RASS level during the EM sessions in Results section.

  1. CAM-ICU and ICDSC was used to identify delirium - please provide details.

Response: We have added the information of identification of delirium as follows: Delirium was monitored, diagnosed and recorded by the trained ICU nurses and/or intensivists every 6 hours or when needed using the Confusion Assessment Method for the ICU or Intensive Care Delirium Screening Checklist, as appropriate.

I have no further comments and wish the authors all the best for their future endeavors.

With best regards

Response: Thank you very much for the important comments.

Round 2

Reviewer 1 Report

2nd Review Letters

Title: Current status of postintensive care syndrome in patients with severe acute respiratory failure caused by coronavirus disease 2019: a retrospective observational cohort study.

Dear Authors,

I saw your corrections in the first review. Some were corrected, some were not.

My first question was “In the author's paper, EM is considered to be an important factor. However, without explaining this important factor in detail, the authors express only as "The EM program was implemented according to the protocol that was published in a previous study (8), with modification." This is thought to be difficult for readers to understand, so please explain in detail the EM program you performed in this study.”

Your response is: “Briefly, the EM program consists of 5 stepwise levels: passive range of motion, head up 30 degrees, sitting on the edge of bed, active transfer to wheelchair, and standing with assistance. If achieving the level without any unfavorable event, the patient proceeds to the next higher level. EM is carried out by either a physician, nurse, physical therapist, or occupational therapist. Each session is 20 minutes.”

The question seemed to have been answered well. But, EM performed on each patient alternately by several people for 20 minutes? Such a case is said to be very unreasonable under experimental conditions. This is because the intervention conditions of several people cannot work the same for each patient.

Second, Has the reliability of the questionnaires used by the authors been investigated?

Third, Data assessed by questionnaires that have not been tested for reliability are discrediting. Because of this, I wonder if the results as well as the discussion were expressed correctly.

I understand to some extent the answers to my second and third questions, but it is regrettable that Cronbach'a should have presented at least for each questionnaire.

I initially said "All references should be formatted to fit the J. Clin. Med.", but unfortunately it doesn't seem to have been fixed at all.

As a final piece of advice, I think you should extract the title from "implementation of an early rehabilitation program plays some role in preventing muscle weakness and that decreasing psychiatric disorders" at the end of your introduction. In addition to what I pointed out in the first review, it was a pity that your thesis used abbreviated words without rule or the types of references did not match with Journal of Clinical Medicine.

I hope my review has helped improve your thesis.

Author Response

Dear Authors,

I saw your corrections in the first review. Some were corrected, some were not.

My first question was “In the author's paper, EM is considered to be an important factor. However, without explaining this important factor in detail, the authors express only as "The EM program was implemented according to the protocol that was published in a previous study (8), with modification." This is thought to be difficult for readers to understand, so please explain in detail the EM program you performed in this study.”

Your response is: “Briefly, the EM program consists of 5 stepwise levels: passive range of motion, head up 30 degrees, sitting on the edge of bed, active transfer to wheelchair, and standing with assistance. If achieving the level without any unfavorable event, the patient proceeds to the next higher level. EM is carried out by either a physician, nurse, physical therapist, or occupational therapist. Each session is 20 minutes.”

The question seemed to have been answered well. But, EM performed on each patient alternately by several people for 20 minutes? Such a case is said to be very unreasonable under experimental conditions. This is because the intervention conditions of several people cannot work the same for each patient.

Response: Thank you for your comment. EM program provided in the study is widely used in the world since it was published in 2008. It is simple and the target of each step is clearly determined so that easy to apply. We have been using it more than 10 years in the ICU. All types of healthcare professionals working in our ICU have been trained and are able to provide. Therefore, although we did not evaluate the inter-staff difference of EM quality, we don’t think it causes big difference on the outcome. The authors have to correct the explanation. Each EM session is 20 min or more. The authors apologize missing words.

Second, Has the reliability of the questionnaires used by the authors been investigated?

Third, Data assessed by questionnaires that have not been tested for reliability are discrediting. Because of this, I wonder if the results as well as the discussion were expressed correctly.

I understand to some extent the answers to my second and third questions, but it is regrettable that Cronbach'a should have presented at least for each questionnaire.

Response: Thank you for pointing out the very important issue. The authors apologize that we did not understand the first question correctly. The questioners are two rehabilitation physician who have been training to use the questionnaire. The intra-investigator reliability is confirmed. Eleven of the 13 patients were asked questions by the senior doctor, and the remaining two were asked by another doctor because the senior doctor was unavailable. We believe that the results were reliable based on our experience so far, but since we did not evaluate inter-investigator reliability in this study, we added it as a study limitation. The authors have also added the sentences in Methods section to explain how to carry out the questionnaire.

I initially said "All references should be formatted to fit the J. Clin. Med.", but unfortunately it doesn't seem to have been fixed at all.

As a final piece of advice, I think you should extract the title from "implementation of an early rehabilitation program plays some role in preventing muscle weakness and that decreasing psychiatric disorders" at the end of your introduction. In addition to what I pointed out in the first review, it was a pity that your thesis used abbreviated words without rule or the types of references did not match with Journal of Clinical Medicine.

I hope my review has helped improve your thesis.

Response: The authors really appreciate the advises which improve our manuscript.

According to the reviewer comment we have changed the title as follows:

Roles of early mobilization program in preventing muscle weakness and decreasing psychiatric disorders in patients with coronavirus disease 2019 pneumonia: a retrospective observational cohort study.